# PREDICTING THE IMPACT OF DATASET COMPOSITION ON MODEL PERFORMANCE

## ABSTRACT

Real-world machine learning systems are often are trained using a mix of data sources with varying cost and quality. Understanding how the size and composition of a training dataset affect model performance is critical for advancing our understanding of generalization, as well as designing more effective data collection policies. We show that there is a simple, accurate way to predict the loss incurred by a model based on data size and composition. Our work expands recent observations of log-linear generalization error and uses this to cast model performance prediction as a learning problem. Using the theory of optimal experimental design, we derive a simple rational function approximation to generalization error that can be fitted using a few model training runs. Our approach achieves nearly exact ($r^2 > .93$) predictions of model performance under substantial extrapolation in two different standard supervised learning tasks and is accurate ($r^2 > .83$) on more challenging machine translation and question answering tasks where baselines achieve worse-than-random performance.

## 1 INTRODUCTION

The success of large scale machine learning systems depends critically on the quantity and quality of data used during training, and we cannot expect these systems to succeed if there is not enough training data or if that data does not cover all the phenomena contained in the test distribution (Ben-David et al., 2010). Knowing this, the designer of a machine learning system might create multiple sources of data, with each one targeting a different feature or domain that the model ought to do well on (Crammer et al., 2007; Wang et al., 2019a). This data-driven design strategy provides powerful tools to improve and evaluate model behavior, but also poses an additional challenge: what is the right way to combine these various data sources? What is the optimal data collection policy for a given budget?

Our goal is to answer these questions by quantifying the relationship between data sources and model performance – how well will our model do if we were to train it on $n$ samples using a data mixture $(q_1 \ldots q_k)$ over our $K$ data sources. A precise model for predicting model performance will allow us to both identify the optimal data collection policy and quantify cost-performance tradeoffs.

The starting point of our work is the recent observation across speech, vision and text (Hestness et al., 2017; Kaplan et al., 2020; Rosenfeld et al., 2020) that the empirical performance of a model is remarkably predictable, and follows the log-linear formula

$$\log(\text{error}) \approx -\alpha \log(n) + C. \tag{1}$$

In this work, we expand this observation to the multi-data-source setting and discover the surprising fact that the slope of the log-linear relationship ($\alpha$) does not vary with data composition and that the data composition only affects the intercept ($C$).

The simple dependence of log-error on data size allows us to reduce the problem of estimating model error into a learning problem. Our approach is straightforward: we hypothesize that model error follows $V(n, q) := \exp(-\alpha \log(\text{n}) + \log(C(q)))$ for a simple parametric functional form $C(q)$, and fit this to observed pairs of $(n, q, \text{error})$ that we obtain by subsampling the dataset and re-training a model. We show that there is a natural and simple choice of $C(q)$ as a rational function that we derive from optimal experimental design for linear regression, M-estimation, and nonparametric

smoothing. The simple and parametric dependence of $V(n, q)$ on $n$ allows us to use our resulting estimates to predict model performance under substantial extrapolation in data size.

Empirically, the resulting predictions are extremely accurate and hold under substantial extrapolation. On the Amazon review prediction dataset (Mansour et al., 2009), we can learn to predict model performance nearly perfectly ($r^2 = 0.96$) from a small dataset of 1200 examples across 3 sources and extrapolate to predict the model error on datasets of up to 4000 examples. We show this high accuracy continues to hold on a real-world task oriented dialogue system ($r^2 = 0.93$), a multi-domain machine translation system ($r^2 = 0.83$), and boolean question answering with weak supervision ($r^2 = 0.86$). In each of the cases, our proposed approach substantially outperforms the best baseline, with the baselines performing worse-than-random in both the machine translation and question answering tasks.

**Related work**  Quantifying the effect of data composition on model performance is closely related to the classical ideas of optimal experimental design, as well as more recent machine learning methods such as active learning and data valuation.

Our work will draw inspiration from the classical $V$-optimal experimental design (John & Draper, 1975) as a way to understand how model performance will change with the data collection policies. However, our approach differs substantially beyond this. Instead of making strong linearity assumptions and identifying closed form formulas for model performance, we treat identifying the impact of data sources on errors as itself a prediction problem, which allows us to quantify these effects for neural networks and non-separable objectives.

Active learning provides methods for incrementally selecting new points to rapidly reduce a loss (Hanneke, 2007). These approaches only consider the problem of optimal data collection and do not seek to predict model performance under *all* data collection strategies (including suboptimal ones), which is critical when making cost-performance tradeoffs across data sources. The model performance predictions produced in our work complements existing work on active learning by providing accurate forecasts of model performance under different data collection strategies.

Finally, data valuation methods such as the Shapley value attempt to assign estimate the impact of a data source on model performance (Ghorbani & Zou, 2019; Jia et al., 2019; Ghorbani et al., 2020; Yoon et al., 2019). These approaches are natural when pricing data sources as part of a market mechanism (Ohrimenko et al., 2019; Agarwal et al., 2019) due to the axiomatic properties of the Shapley value. Our approach differs in that we seek simply to estimate the performance of a model rather than to assign a single price to examples from a data source. This difference means that axioms such as *additivity* that are critical for the Shapley value are not relevant for our goal. We show that for the purpose of predicting errors, a rational function (rather than a linear cost) follows naturally from optimal experimental design. Our experiments also show that our rational function approximation provides better model performance predictions than a linear, additive model.

## 2 PROBLEM STATEMENT AND EMPIRICAL OBSERVATIONS

Our goal is to predict the performance of a model as a function of the number of training samples $n$ as well as the dataset composition $q$, where $q_k$ represents the fraction of the training data drawn from data source $k$. We will now define this goal more formally in terms of the training data distribution, model fitting, and test loss.

The training data consists of an $n$-sample training set $p_{n,q}$ that is created by sampling from the mixture $p := \sum_{k \in [K]} q_k p_k$ where $p_k$ are data generating distributions for each of the $K$ data sources and $q_k$ are mixture weights with $q_k \geq 0$ and $\sum_{k \in [K]} q_k = 1$. Using this dataset, we learn a prediction model $\hat{\theta}$ that incurs loss $\ell(\hat{\theta}; x, y)$ for a training example $(x, y)$. The fitted model is the empirical loss minimizer, which we define as

$$\hat{\theta}(p_{n,q}) := \arg\min_{\theta \in \Theta} \mathbb{E}_{p_{n,q}} \left[ \ell(\theta; x, y) \right].$$

The performance of this classifier is evaluated on a test distribution which may differ from the training distribution by a covariate shift (i.e. $p(y \mid x) = p_{\text{test}}(y \mid x)$). We are interested in model performance as a function of the data size and composition (and not a fixed empirical distribution

$p_{n,q}$) and thus our goal is to predict the model's expected excess loss over draws in both the training and test distributions,

$$L(n, q) := \mathbb{E}\left[\ell(\hat{\theta}(p_{n,q}); x, y)\right] - \inf_{\theta} \mathbb{E}\left[\ell(\theta; x, y)\right].$$

Estimating $L$ requires that we hypothesize a relationship between $(n, q)$ and the expected model loss. Following earlier observations by Hestness et al. (2017), we expect a log-linear relationship between $L(n, q)$ and $\log(n)$ for any fixed $q$, which implies a possible approximation as

$$\log(L(n, q)) \approx \log(V(n, q)) := \alpha(q)\log(n) + C(q). \tag{2}$$

We now examine this hypothesis in a simple toy example.

**Linear toy data:** We will start with the simplest nontrivial example of linear least-squares regression to study $L(n, q)$. In this example, there are two data sources over $x \in \mathbb{R}^2$. The first data source has substantial variability on the first coordinate $x_0$ but not $x_1$ and vice versa for the second data source. The overall generative process is

$$y \mid x \sim [0.5, 1]^\top x + \epsilon \qquad z \sim \text{Bern}(q) \qquad \epsilon \sim N(0, 1)$$

$$x \mid z = 0 \sim N\left(0, \begin{bmatrix} 1 & 0 \\ 0 & 0.001 \end{bmatrix}\right) \qquad x \mid z = 1 \sim N\left(0, \begin{bmatrix} 0.001 & 0 \\ 0 & 1 \end{bmatrix}\right).$$

Let $L(n, q)$ be the excess squared loss of a linear least squares model trained with $n$ samples from a mixture $q$ and evaluated on a test distribution with $q = 0.5$. What will $L(n, q)$ look like? Figure 1a shows a clear linear relationship between log dataset size $(\log(n))$ and $\log(L(n, q))$. The intercept of the linear relationship seems to vary with the data mixture $q$, but the slope seems constant.

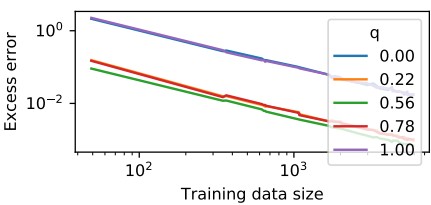
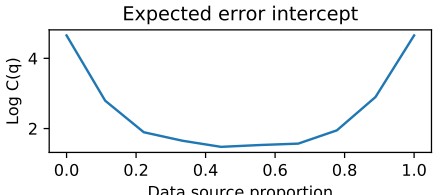

(a) Log excess loss (y axis) is linear with log-dataset size (x axis). Changing the data distribution by varying $q$ (line color) changes the intercept but not the slope.

(b) Intercept ($C(q)$) of the loss-dataset log linear relationship. The loss is lowest when the dataset is a mix of both data sources ($q \approx 0.5$) and rapidly increases when exclusively using one data source.

**Figure 1:** The log-linear effects of data composition and size on the linear toy dataset.

Examining Figure 1a more closely, we find that the extremes of using either data source exclusively (blue / purple lines) performs worse than a mix suggesting that $\log(L(n, q))$ is unlikely to be linear in $q$. Intuitively, we can think of each data distribution as having a different strength (i.e. more variance in either $x_0$ or $x_1$) and combining the two results in a better data distribution than either alone. We can see this more clearly when we estimate the intercept for each of these lines (Figure 1b). The estimated intercepts show a U-shaped curve that rapidly increases as $q \to 0$ or $q \to 1$ and is generally flat from 0.2 to 0.8.

## 3 METHOD AND THEORY

We have observed that in the case of a simple linear regression, the log-error not only follows the relationship outlined in equation 2, but also that the slope $\alpha$ is constant as we vary the data composition (and we will further validate this claim on more complex tasks and models in subsequent sections). This observation shows we may be able to further simplify the log-linear approximation as

$$\log(L(n, q)) \approx \log(V(n, q)) := -\alpha\log(n) + \log(C(q)).$$

Now note that this functional form decouples the data size $n$ and mixture proportions $C(q)$ into two terms. This is the key observation of our work: $\log(V(n, q))$ has a very simple dependence on $n$,

and the more complex term $C(q)$ has no dependence on $n$. Therefore we can cast this as a learning problem, where we learn $\alpha$ and a parametric function $C_\lambda(q)$ based on the model's error over a range of $q$ and small $n$, and extrapolate this for large $n$ using the log-linear dependence of $\log V$ on $n$.

Concretely, given a dataset with $\{n_1 \ldots n_k\}$ we can generate a subsampled dataset with $\hat{n}_k \sim$ Unif$(0, n_k)$ samples from each source. This results in a training set with data size $\hat{n} = \sum_k \hat{n}_k$ and composition $\hat{q}_k = \frac{\hat{n}_k}{\hat{n}}$. We fit a model to this subsampled data and compute its loss $L(\hat{n}, \hat{q})$. Given the triple $(\hat{n}, \hat{q}, L(\hat{n}, \hat{q}))$ we can now simply fit the hypothesized functional form,

$$\min_{\lambda, \alpha} \mathbb{E}_{\hat{q}, \hat{n}} \left[ \left( \log(L(\hat{n}, \hat{q})) - \alpha \log(\hat{n}) + \log(C_\lambda(\hat{q})) \right)^2 \right].$$

The experimental data does not specify the functional form of $C_\lambda(q)$ except that it should handle convex functions like those seen in Figure 1b. We will now study $V(n, q)$ theoretically and argue that a natural choice is the rational function

$$C_\lambda(q) := \sum_{i=1}^{M} \left( \sum_{k=1}^{K} \lambda_{ik} q_k \right)^{-1}.$$

In the subsequent sections, we will study three settings: ordinary linear regression, M-estimation, and nonparametric regression and show that our hypothesized log-linear approximation arises naturally in all three cases.

## 3.1 Linear regression

We begin by characterizing $L(n, q)$ in the linear regression case, where we can derive closed form expressions for the expected loss as a function of training data. Our setting is $d$-dimensional, $n$-sample linear regression, defined as $y = x^\top \beta + \epsilon$ with i.i.d. $\epsilon \sim N(0, 1)$. Our training data follows $x \sim p := \sum_{k \in [K]} q_k p_k$ where each data source has full-rank second moments $\Sigma_k := \mathbb{E}_{x \sim p_k} \left[ xx^\top \right]$.

Define the ordinary least squares estimator $\hat{\beta} := (X^\top X)^{-1} X^\top Y$ in terms of the features $X \in \mathbb{R}^{n \times d}$ and $Y \in \mathbb{R}^n$. The excess test loss of this estimator over any $x^* \sim p^*$ and $y^* := x^{*\top} \beta + \epsilon$ is defined as

$$L(n, q) = \mathbb{E}[\|x^*(\beta - \hat{\beta})\|_2^2].$$

The theory of V-optimal experimental design(Pukelsheim, 2006) allows us to characterize this excess loss.

**Proposition 3.1.** *The excess expected loss for ordinary least squares trained on a mixture $q$ with data size $n$ and subgaussian $x$ follows*

$$\log(L(n, q)) = -\log(n) + \log \left( \underbrace{Tr \left( \Sigma^* \left( \sum_{k \in [K]} q_k \Sigma_k \right)^{-1} \right)}_{C(q)} \right) + O \left( \frac{\sqrt{\log(1/\delta)}}{\sqrt{n}} \right),$$

*with probability at least $1 - \delta$ where $\Sigma^* := \mathbb{E}_{x \sim p^*} \left[ xx^\top \right]$ and $\Sigma_k := \mathbb{E}_{x \sim p_k} \left[ xx^\top \right]$.*

We will defer all proofs to the appendix due to space constraints. Clearly $C(q)$ is not linear even in this simple case, and the terms for $q_k$ appear within an inverse. Naively, we might hypothesize that it behaves much more closely to a linear rational function (i.e. $(\sum_i \lambda_i q_i)^{-1}$) and this intuition will turn out to be correct whenever $\Sigma^*$ and $\Sigma_k$ are approximately diagonalizable.

**Corollary 3.1.** *Let $P$ be an orthogonal matrix which approximately simultaneously diagonalizes $P^{-1} \Sigma^* P = D^*$, $P^{-1} \Sigma_k P = D_k + R_k$ for diagonal some matrices $D$. Then for full-rank $\Sigma^*$ and sufficiently small $R_k$,*

$$Tr \left( \Sigma^* \left( \sum_{k \in [K]} q_k \Sigma_k \right)^{-1} \right) = \sum_{i \in [d]} \frac{D_{ii}^*}{\sum_k q_k D_{k,ii}} + o \left( \| \sum_k q_k R_k \|_F \right).$$

The first order term exactly matches the hypothesized $C(q)$ as a rational function with $d$ terms and validates this choice for linear regression. To interpret this corollary, the approximate diagonalizability condition states that the eigenvectors for $\Sigma^*$ and $\Sigma_k$ coincide, and that $D_{ii}^*$ and $D_{k,ii}$ are these eigenvalues. The ratio $\frac{D_{ii}^*}{\sum_k q_k D_{k,ii}}$ measures the ratio of variance in the test distribution to that of the training distribution for the $i$-th eigenvector.

The key observation is that the variance (i.e. the information each data source contributes to a particular coordinate $i$) is linear, but the dependence of model error to training variance is *inverse* and that there are $d$ different coordinates making the overall dependence of errors on data composition nonlinear. There are clear qualitative differences between a linear and rational function approximation to $C(q)$, with the rational function being strongly convex with diminishing returns in $q$.

## 3.2 General M estimators

We might rightfully ask whether this kind of approximation continues to hold for nonlinear models and losses like neural networks. The same analysis as above can be extended to the asymptotic behavior of a substantially more general class of models known as $M$-estimators, which are empirical loss minimizers of a differentiable loss.

For the regression case, we relied on a closed-form characterization of $\beta$. For M-estimators we will use asymptotic normality under the sampling distribution,

**Theorem 3.1** (van der Vaart (1998)). *Consider a twice differentiable loss $\ell$ whose gradients are bounded and Donsker. Let $\theta_n$ be an estimator which fulfills the approximate first-order optimality condition with minimizer $\theta_\infty$,*

$$\mathbb{E}_{p_n}[\nabla \ell(y, x; \theta_n)] = o(n^{-1/2}) \qquad and \qquad \mathbb{E}_p[\nabla \ell(y, x; \theta_\infty)] = 0.$$

*If $\theta_n \xrightarrow{p} \theta_\infty$ and both $I_{\theta_\infty}^{-1} := \mathbb{E}_p[H\ell(y, x; \theta_\infty)]^{-1}$ and $\Sigma_{\theta_\infty} := \mathbb{E}_p[\nabla \ell(y, x; \theta_\infty)\nabla \ell(y, x; \theta_\infty)^\top]$ exist,*

$$\sqrt{n}(\theta_n - \theta_\infty) \to N\left(0, I_{\theta_\infty}^{-1}\Sigma_{\theta_\infty} I_{\theta_\infty}^{-1}\right).$$

Now that we have the asymptotic distribution of the $M$-estimator, we can quantify the (asymptotic) form of $C(q)$ with respect to a test distribution $p^*$ simply by taking the Taylor expansion of the loss at $\theta_\infty$.

**Corollary 3.2.** *Under the conditions of Theorem 3.1, let $\ell(y, x; \theta) = -\log p_\theta(y \mid x)$ and there exists some $\theta^* = \theta_\infty$ such that $p_{\theta^*}(y \mid x) = p(y \mid x)$ then*

$$\log(L(n, q)) = -\log(n) + \log\left(Tr\left(\Sigma^*\left(\sum_k q_k \Sigma_k\right)^{-1}\right) + o(n^{-1})\right).$$

*for $\Sigma_k := \mathbb{E}_{p_k}[H\ell(y, x; \theta^*)]$ and $\Sigma^* := \mathbb{E}_{p^*}[H\ell(y, x; \theta^*)]$*

This result relies on two additional assumptions: the loss is a log loss, and the model is well-specified. The first assumption is weak, as many models today use log softmax type losses. The well-specified assumption is stronger but may be reasonable for nearly nonparametric functions such as neural networks. For a less simple but more general result, see Corollary A.1 in the appendix.

This has the same functional form as before: $C(q)$ is the trace of a test distribution dependent matrix $\Sigma^*$ and the inverse of data source matrices $\Sigma_k$. The difference now is that instead of covariances, we are looking at the Hessian of the parameters with respect to the unknown optimal model $\theta^*$. Applying the simultaneous diagonalization argument from earlier once again results in a rational function that is captured by $C(q)$.

## 3.3 Nonparametric models

Finally, we show that the same relationship holds for nonparametric models such as kernel smoothing or binning. Our goal will be to estimate some ground truth map $y = f(x) + \epsilon$ for $\epsilon$ i.i.d $N(0, 1)$ and $f$ a differentiable $L$-Lipschitz function. The quality of an estimate will be measured by some twice-differentiable loss $\ell(y, x)$ with bounded first two derivatives.

Given $n$ samples $(x_1, y_1) \ldots (x_n, y_n) \in [0,1]^d \times \mathbb{R}$ drawn i.i.d from some density $p = \sum_k q_k p_k$, one natural estimator for this problem is the nonparametric binning estimator $\hat{f}$ which we define in terms of axis-aligned hypercubes $B_\delta(x, S) := \{x' \in S : \lfloor x'/\delta \rfloor = \lfloor x/\delta \rfloor\}$. Let $X_n := \{x_1 \ldots x_n\}$ then we can define our estimator,

$$\hat{f}_\delta(x) := \frac{1}{|B_\delta(x, X_n)|} \sum_{x_i \in B_\delta(x, X_n)} y_i.$$

Assuming we choose $\delta$ and $n$ sufficiently large that each bin concentrates to its expected value, we have the following error estimate

**Proposition 3.2.** *Let $B_\delta(x, p_k) = \mathbb{E}_{x' \sim p_k}[|B_\delta(x, \{x'\})|]$ be the probability of drawing some $x' \sim p_k$ in the same bin as $x$, and assume $B_\delta(x, p_k)$ is bounded away from zero. Then*

$$L(n, q) := \mathbb{E}[\ell(\hat{f}_\delta(x), x) - \ell(f(x), x)]$$

$$= \mathbb{E}\left[\frac{\ell''(f(x), x)}{\sum_k q_k B_\delta(x, p_k)}\right] + O\left(\frac{\sqrt{\log(\gamma^{-1}) + d\log(\delta)}}{\sqrt{2n}}\right) + O(L\delta\sqrt{d} + L^2\delta^2 d),$$

*holds with probability at least $1 - \gamma$, where the expectation is taken with respect to draws of $y$.*

Once again, we see a rational function in $q$, with no further approximation needed. Each bin is a term in the rational function approximation with weight $\ell''(x)$.

## 4 Experiments

We have seen that a rational function is a reasonable approximation to $C(q)$ across 3 different settings. We will now show that this is the case in practice, and additionally that $C(q)$ can be accurately estimated using a few models trained on small datasets. The resulting estimates of model performance are accurate for models with an order of magnitude more data.

**Baselines and implementation**   Our evaluations focus on our ability to predict the loss incurred by a model $L(n, q)$. To do so, we will compare the rational function approximation procedure against several natural baselines for predicting the loss of a model. Each of the baselines correspond to a different assumption about the functional form of $\log(V(n, q))$ that we use to approximate $\log(L(n, q))$.

**Datasize**: Assume a functional form of $\log(V(n, q)) = \alpha \log(n) + c$ ignoring the data composition and dependence on $q$.

**Linear**: Assume a functional form of $\log(V(n, q)) = \alpha \log(n) + \beta^\top q + c$. This is the natural approach if we treat $\log(V(n, q))$ as linear in $q$ and log-linear in $n$.

**Ablation** and **Shapley**: further constrain the linear baseline by setting $\beta$ to either the log-Shapley value obtained as the marginal contribution of a data source (for the Shapley baselines) or the log-ratio of losses obtained after removing a data source (ablation). We use this approach as we found it to dominate the usual assumption of treating $V(n, q)$ as being linear in the Shapley value.

As the baselines are all linear in $\log(n)$ and $q$ we solve the optimal parameters for these models in closed form with least squares regression. We will refer to our approach as **Rational**, and we fit this using the Adagrad (Duchi et al., 2010) optimizer with 20000 steps and learning rate set via cross validation over the interval [0.005, 0.5]. The number of terms in the rational function sum is set to be one more than the number of data sources in all experiments.

### 4.1 Focused evaluation: Amazon sentiment

We now consider the Amazon sentiment prediction dataset in (Mansour et al., 2009) where the goal is to predict Amazon ratings for books (from 0 to 5 stars) using bag-of-words features from the reviews. The training data comes from 3 domains that differ from the test data: kitchen, DVD, and electronics reviews. The model is a standard ridge regularized regression model. Our experimental setup for estimating model loss is the following: we uniformly randomly sample the dataset size for

each source (resulting in between 0 and 1200 examples for each source), and train a model on this dataset. We measure the test error via average squared loss on the books domain.

We fit $V(n, q)$ with 4 terms for $C(q)$ by minimizing the squared loss with respect to log-error on models containing 0-1200 examples total. We then use $V(n, q)$ to predict log-error on the models trained on 1200-3600 examples from each domain. The results of this extrapolation task are shown in Table 2. Our $V(n, q)$ estimate is nearly perfect ($r^2 = 0.96$) and extrapolate from the low data to high data regime without issue. This correlation is substantially higher than either using data set size ($r^2 = -0.65$), a linear model ($r^2 = 0.76$) and even better *the training error* of the best additive model ($r^2 = 0.87$).

| Metric | Datasize | Ablation | Shapley | Linear | Rational |
|---|---|---|---|---|---|
| Train | 0.20 | 0.77 | 0.80 | 0.87 | 0.95 |
| Extrapolation | -0.65 | 0.43 | 0.51 | 0.76 | **0.96** |
| Bootstrap interval | (-1.86, -0.08) | (0.018, 0.60) | (0.15, 0.66) | (0.58, 0.83) | (0.94, 0.97) |

**Table 1.** Accuracy of $L(n, q)$ estimates on the Amazon review sentiment prediction task. Bold indicates the best performing model under extrapolation, identified by a bootstrapped paired difference test.

The data size predictor has a negative $r^2$ on the extrapolation setting which may seem surprising. However, this can happen whenever a predictor fails to perform better than predicting the mean of the test set. It is nontrivial to predict the mean of the test set in an extrapolation setting, and in this case, data size estimates are generally uninformative as data from the kitchen domain is not useful for predicting book review scores.

## 4.2 Broad evaluation: semantic parsing, translation, and question answering

We now perform a broader evaluation of the 3 methods (linear, rational, and datasize) on 3 tasks that violate our assumptions about model performance prediction. We excluded the two ablation based methods as they are special cases of the linear model, and generally performed worse.

| method | Task-oriented dialogue | | Machine Translation | | Multitask QA | |
|---|---|---|---|---|---|---|
| | Train | Extrapolation | Train | Extrapolation | Train | Extrapolation |
| Datasize | 0.54 | 0.69 (0.52, 0.81) | 0.05 | -0.80 (-3.37, -0.03) | 0.78 | -1.39 (-6.4, -0.88) |
| Linear | 0.99 | 0.91 (0.78, 0.96) | 0.27 | -0.69 (-3.10, -0.03) | 0.98 | -0.92 (-6.63, -0.17) |
| Rational | 0.99 | **0.93** (0.83, 0.97) | 0.98 | **0.83** (0.59, 0.92) | 0.97 | **0.86** (0.66, 0.89) |

**Table 2.** Accuracy of error estimates on 3 real-world tasks that pose challenges for performance prediction due to their use of deep neural networks, non-separable losses such as BLEU, and weak supervision. In all cases, the rational function approximation provides good estimates of model performance.

**Task-oriented dialogue**  We perform this analysis on a real world task-oriented dialogue system based upon the SMCalFlow dataset and model (Andreas et al., 2020). The task differs from the Amazon setting in two ways: the model is a nonlinear neural model for which there is no closed form optimal experimental design and the task is semantic parsing which is a more complex structured prediction problem. There are 105727 total dialogues across 4 data sources consisting of a wizard-of-oz style crowdsourced dialogues, paraphrases of existing dialogues, on-policy dialogues between the system and crowdworkers, and hand-crafted dialogues by expert data scientists. We sample the number of dialogues for each source with a uniform distribution to determine $q$ and then further subsample each data source by $[0.1, 0.3, 0.7, 1.0]$ to vary $n$. Test errors are measured by whether the execution of the model matches human references.

We fit $V(n, q)$ with 5 terms for $C(q)$ on 10 models containing less than 16,000 examples, and testing on 19 models containing between 16,000 and 100,000 examples. The results in Table 2 show our approach is accurate ($r^2 = 0.93$) and outperforms baselines including data size ($r^2 = 0.69$) and the additive model ($r^2 = 0.91$). While the bootstrap intervals for the $r^2$ for the rational function approximation is sufficiently wide that it contains the mean $r^2$ estimate of the linear model, a more powerful paired difference test between the linear and rational approximations shows that this gap is statistically significant at a 5% level for the bootstrap distribution.

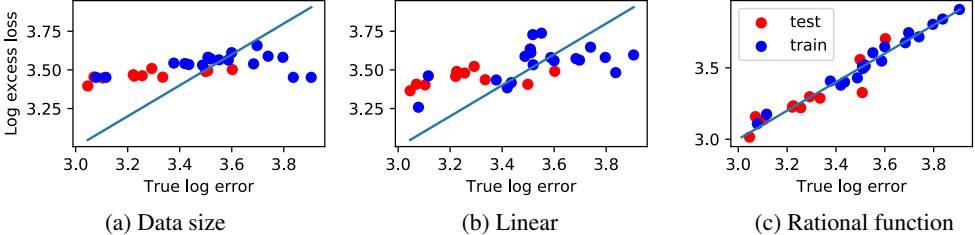

**Figure 2.** Performance prediction on a multi-domain machine translation task with BLEU as the performance measure. There is little correlation between dataset size and loss (left panel) while the rational function approximation provides reasonable predictions (right).

**Machine translation** Thus far, we have evaluated on separable losses such as mean squared error, or model accuracy. We now show that our approach to predicting model performance continues to work for non-separable losses such as BLEU for machine translation. Our task is the standard multi-domain machine translation dataset from Koehn & Knowles (2017). We use the preprocessed data, model, and hyperparameters from Hu et al. (2019) for a baseline sequence to sequence machine translation model. The model is trained on 4 data sources: Acquis (legal text), EMEA (parliamentary proceedings), IT (IT assistance), and Koran (translations of the Quran). Evaluation is performed on the Acquis test set using sacrebleu to compute BLUE (Post, 2018).

To estimate the performance of models under varying data composition, we subsample up to 300,000 sentences from each data source, fit the estimators on 19 datasets of size less than 600,000 total sentences, and evaluate on 11 datasets of size 600,000 to 1,200,000. Since BLEU is a similarity measure and is penalized by reference ambiguity, we consider 50-BLEU to be the excess error. The rational function approximation is the only procedure to achieve a positive $r^2$ (0.83) among the baselines. The difference in prediction accuracies is apparent when plotting predicted and observed log-loss (Figure 2). The linear model even has low *training* set $r^2$, suggesting that the relationship between data composition and performance is fundamentally nonlinear.

**Multitask question answering** Finally, we consider a multitask learning problem where some of the data sources are auxiliary tasks that may not directly be useful for the test time task. This breaks the covariate shift assumption that has been implicit throughout this paper. The target task is the BoolQ question answering dataset , and we train this model using a combination of 4 data sources: the MNLI entailment task (Williams et al. (2018), 50,000 examples subsampled), STS sentence similarity judgment task (Cer et al. (2017), 5749 examples), MRPC paraphrasing task (Dolan et al. (2004), 3668 examples), and the BoolQ training set (Clark et al. (2019) 9427 examples). We use the GLUE data with the Jiant package to train a baseline BERT based model for this task (Wang et al., 2019b).

The challenge with this task is that only the BoolQ training set provides direct supervision for the test-time task, and the other data sources provide weak supervision that may or may not be helpful in the downstream problem. The model performance estimates are fitted on 6 datasets with up to 25,000 total examples and evaluated on 17 datasets with more than 25,000 examples. As expected, the data size based performance estimates are catastrophically bad, resulting in negative correlations. The linear estimates do not seem to extrapolate well to the test set. The rational function approximation is the only one of the three to provide positive $r^2$ on this task.

## 5  DISCUSSION

In this work, we've proposed a new approach to predicting the performance of a prediction model as a function of training data composition that consists of measuring model accuracies for small $n$ and a range of $q$ and fitting a parametric model $V(n, q) := -\alpha \log(n) + \sum_{i=1}^{m} (\sum_{k=1}^{K} \lambda_{ik} q_k)^{-1}$. We show that this parametric model is a natural approximation to model performance for a range of models, and accurately predicts the empirical performance of models in an extrapolation setting. Our work is the first step in going beyond closed-form estimates of model performance or additivity assumptions. It is an open question whether the same approach can scale to large numbers of data sources, and we hope to explore this in future work.

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

## A    APPENDIX

**Proposition 3.1.** *The excess expected loss for ordinary least squares trained on a mixture q with data size n and subgaussian x follows*

$$
\log(L(n, q)) = -\log(n) + \log \left( \underbrace{Tr \left( \Sigma^* \left( \sum_{k \in [K]} q_k \Sigma_k \right)^{-1} \right)}_{C(q)} \right) + O \left( \frac{\sqrt{\log(1/\delta)}}{\sqrt{n}} \right),
$$

*with probability at least $1 - \delta$ where $\Sigma^* := \mathbb{E}_{x \sim p^*} \left[ xx^\top \right]$ and $\Sigma_k := \mathbb{E}_{x \sim p_k} \left[ xx^\top \right]$.*

**Proof**    We will begin by deriving the excess loss for a fixed set of test examples $X^* \in \mathbb{R}^{m \times d}$ and training examples $X \in \mathbb{R}^{n \times d}$. We are interested in the excess loss, which can be written in a simple form due to strict exogeniety of the least squares regression,

$$
\|X^*(\beta - \hat{\beta})\|_2^2 / m = \mathrm{Tr}(X^{*\top} X^* \left( X^\top X \right)^{-1}) / m.
$$

This is a classic instance of V-optimal design. We reproduce this result for clarity. Let $e := (\beta - \hat{\beta})$ then,

$$
\|X^*(\beta - \hat{\beta})\|_2^2 / m = \mathrm{Tr}(e^\top X^{\top *} X^* e) / m = \mathrm{Tr}(X^{*\top} X^* \left( X^\top X \right)^{-1}) / m.
$$

The challenge now is to bound the expected loss,

$$
L(n, q) = \mathbb{E} \left[ \mathrm{Tr} \left( X^{*\top} X^* \left( X^\top X \right)^{-1} \right) \right] / m.
$$

The expectation with respect to $X^*$ is straightforward, but the expectation with respect to $X$ is challenging as $\mathbb{E}[X^\top X]^{-1} \neq \mathbb{E}[(X^\top X)^{-1}]$. Since $x$ is subgaussian, we can make use of matrix concentration to show this term concentrates at appropriate rate. Let $\Sigma := \mathbb{E}[X^\top X]$ then by Vershynin (2010, Corollary 5.50) for sufficiently large $n$,

$$
\|X^\top X n^{-1} - \Sigma\|_{op} \leq C \frac{\sqrt{\log(1/\delta)}}{\sqrt{n}}
$$

with probability $1 - \delta$ and constant $C$ depending on the subgaussianity parameter of $X$. Now we can expand the empirical covariance in terms of the expectation and a residual $\Delta = X^\top X / n - \Sigma$ using the identity $(\Sigma + \Delta)^{-1} = \Sigma^{-1} + \sum_{t=1}^{\infty} \Sigma^{-1} (\Delta \Sigma^{-1})^t$ as

$$
L(n, q) = \mathrm{Tr} \left( \Sigma^* \Sigma^{-1} n^{-1} \right) + n^{-1} \mathbb{E} \left[ \mathrm{Tr} \left( \sum_{t=1}^{\infty} \Sigma^{-1} (\Delta \Sigma^{-1})^t \right) \right].
$$

whenever the series is convergent ($\|\Sigma^{-1}\|_{op} > \|\Delta\|_{op}$).

For sufficiently large $n$, this series will converge (as $\sigma_{\min}(\Sigma)$ is a constant and $\|\Delta\|_{op} = O(n^{-1/2})$) and the first term in the series is the dominant one. Using the trace inequality $\mathrm{Tr}(A^{-1} B A^{-1}) \leq d^2 \|A^{-1}\|_{op}^2 \|B\|_{op}$,

$$
L(n, q) = \mathrm{Tr} \left( \Sigma^* \Sigma^{-1} n^{-1} \right) + O \left( \frac{\sqrt{\log(1/\delta)}}{n^{3/2}} \right).
$$

Finally, we make use of the fact that $X$ is drawn from a mixture with component moments $\Sigma_k := \mathbb{E}_{x \sim p_k}[xx^\top]$ and take the Taylor expansion of $\log(L(n, q))$ at $L(n, q) = \mathrm{Tr}(\Sigma^* \Sigma^{-1} n^{-1})$ to obtain,

$$
\log(L(n, q)) = -\log(n) + \log \left( \mathrm{Tr} \left( \Sigma^* \left( \sum_k q_k \Sigma_k \right)^{-1} \right) \right) + O \left( \frac{\sqrt{\log(1/\delta)}}{\sqrt{n}} \right).
$$

with probability $1 - \delta$.    $\square$

**Corollary 3.1.** *Let $P$ be an orthogonal matrix which approximately simultaneously diagonalizes $P^{-1}\Sigma^* P = D^*$, $P^{-1}\Sigma_k P = D_k + R_k$ for diagonal some matrices $D$. Then for full-rank $\Sigma^*$ and sufficiently small $R_k$,*

$$Tr\left(\Sigma^*\left(\sum_{k\in[K]} q_k \Sigma_k\right)^{-1}\right) = \sum_{i\in[d]} \frac{D_{ii}^*}{\sum_k q_k D_{k,ii}} + o\left(\|\sum_k q_k R_k\|_F\right).$$

**Proof** Since $\Sigma^*$ is full rank, we can apply the local expansion $(A + B)^{-1} = A^{-1} + \sum_{i=1}^{\infty} A^{-1}\left(BA^{-1}\right)^i$ along with the trace bound $|Tr(A^{-1}BA^{-1})| \leq \|A^{-1}\|_F^2 \|B\|_F$

$$Tr\left(\Sigma^*(\sum_k \Sigma_k)^{-1}\right) = Tr\left(D^*(\sum_k q_k(D_k + R_k))^{-1}\right)$$

$$= \sum_{i\in[d]} \frac{D_{ii}^*}{\sum_k q_k D_{k,ii}} + o\left(\|\sum_k q_k R_k\|_F\right).$$

Where the last line uses the local expansion and trace bound for sufficiently small $B$. $\qquad\square$

**Theorem 3.1** (van der Vaart (1998)). *Consider a twice differentiable loss $\ell$ whose gradients are bounded and Donsker. Let $\theta_n$ be an estimator which fulfills the approximate first-order optimality condition with minimizer $\theta_\infty$,*

$$\mathbb{E}_{p_n}[\nabla\ell(y, x; \theta_n)] = o(n^{-1/2}) \qquad and \qquad \mathbb{E}_p[\nabla\ell(y, x; \theta_\infty)] = 0.$$

*If $\theta_n \xrightarrow{p} \theta_\infty$ and both $I_{\theta_\infty}^{-1} := \mathbb{E}_p[H\ell(y, x; \theta_\infty)]^{-1}$ and $\Sigma_{\theta_\infty} := \mathbb{E}_p[\nabla\ell(y, x; \theta_\infty)\nabla\ell(y, x; \theta_\infty)^\top]$ exist,*

$$\sqrt{n}(\theta_n - \theta_\infty) \to N\left(0, I_{\theta_\infty}^{-1}\Sigma_{\theta_\infty}I_{\theta_\infty}^{-1}\right).$$

**Proof**

First, we take the first order approximation to the population minimizer

$$\mathbb{E}_p[\nabla\ell(y, x; \theta_n)] = \mathbb{E}_p[\nabla\ell(y, x; \theta_\infty)] + \mathbb{E}_p[H\ell(y, x; \theta_\infty)]^\top(\theta_n - \theta_\infty) + o(\|\theta_n - \theta_\infty\|)$$

Assuming the existence of $I_{\theta_\infty}^{-1}$ and using the approximate first-order optimality conditions for both $\theta_\infty$ and $\theta_n$ we can solve for the parameter difference as

$$\sqrt{n}(\theta_n - \theta_\infty)$$
$$= \sqrt{n}I_{\theta_\infty}^{-1}\left(\mathbb{E}_p[\nabla\ell(y, x; \theta_n)] - \mathbb{E}_p[\nabla\ell(y, x; \theta_\infty)]\right) + o(\sqrt{n}\|\theta_n - \theta_\infty\|)$$
$$= \sqrt{n}I_{\theta_\infty}^{-1}\left(\mathbb{E}_p[\nabla\ell(y, x; \theta_n)] - \mathbb{E}_{p_n}[\nabla\ell(y, x; \theta_n)]\right) + o(1 + \sqrt{n}\|\theta_n - \theta_\infty\|)$$

Now since $\nabla\ell$ is Donsker over each coordinate and $\theta_n \to \theta_\infty$ in probability, we can obtain uniform concentration on the gradients (van der Vaart, 1998)

$$\sqrt{n}\left(\mathbb{E}_p[\nabla\ell(y, x; \theta_n)] - \mathbb{E}_{p_n}[\nabla\ell(y, x; \theta_n)] - \mathbb{E}_p[\nabla\ell(y, x; \theta_\infty)] + \mathbb{E}_{p_n}[\nabla\ell(y, x; \theta_\infty)]\right) = o(1 + \sqrt{n}\|\theta_n - \theta_\infty\|).$$

Substituting this into the earlier equality allows us to replace $\theta_n$ with $\theta_\infty$,

$$\sqrt{n}(\theta_n - \theta_\infty) = \sqrt{n}I_{\theta_\infty}^{-1}\left(\mathbb{E}_{p_n}[\nabla\ell(y, x; \theta_\infty)] - \mathbb{E}_p[\nabla\ell(y, x; \theta_\infty)]\right) + o(1 + \sqrt{n}\|\theta_n - \theta_\infty\|).$$

Since $\nabla\ell$ is bounded and thus has finite third moments, $\mathbb{E}_{p_n}[\nabla\ell(y, x; \theta_\infty)]$ obeys the central limit theorem with distribution $N(\mathbb{E}_p[\nabla\ell(y, x; \theta_\infty)], \Sigma_{\theta_\infty})$. Finally, the lower order terms on the right asymptotically vanish since $\theta_n \to \theta_\infty$ and we obtain the stated result. $\qquad\square$

**Corollary A.1.** *Under the conditions of Theorem 3.1 and either $\mathbb{E}_{p^*}[\nabla\ell(y, x; \theta_\infty)] = 0$ or $\mathbb{E}[\theta_n] = \theta_\infty + o(n^{-1})$,*

$$L(n, q) := \mathbb{E}[\ell(y, x; \theta_n)] - \mathbb{E}[\ell(y, x; \theta_\infty)] = n^{-1}Tr\left(\mathbb{E}_{p^*}[H\ell(y, x; \theta_\infty)]I_{\theta_\infty}^{-1}\Sigma_{\theta_\infty}I_{\theta_\infty}^{-1}\right) + o(n^{-1}).$$

**Proof**  Taking a second order Taylor expansion,

$$\mathbb{E}[\mathbb{E}_{p^*}[\ell(y, x; \theta_n)]] = \mathbb{E}_{p^*}[\ell(y, x; \theta_\infty)]$$
$$+ \mathbb{E}[\mathbb{E}_{p^*}[\nabla \ell(y, x; \theta_\infty)](\theta_\infty - \theta_n)^\top]$$
$$+ \mathbb{E}[\mathbb{E}_{p^*}[(\theta_\infty - \theta_n)]H\ell(y, x; \theta_\infty)(\theta_\infty - \theta_n)^\top] + o(n^{-1})$$

The first-order term is at most $o(n^{-1})$ by the additional assumption. Either our asymptotic parameter estimate is also a stationary point for the test distribution and $\mathbb{E}_{p^*}[\nabla \ell(y, x; \theta_\infty)] = 0$, or our estimator is unbiased and $\mathbb{E}[\theta_\infty - \theta_n] = o(n^{-1})$. The second order term can be simplified via the asymptotic normality of $\theta_n$ as

$$\mathbb{E}[\mathbb{E}_{p^*}[(\theta_\infty - \theta_n)H\ell(y, x; \theta_\infty)(\theta_\infty - \theta_n)^\top]] = n^{-1}\mathrm{Tr}\left(\mathbb{E}_{p^*}[H\ell(y, x; \theta_\infty)]I_{\theta_\infty}^{-1}\Sigma_{\theta_\infty}I_{\theta_\infty}^{-1}\right) + o(n^{-1}).$$

$\square$

**Corollary 3.2.** *Under the conditions of Theorem 3.1, let $\ell(y, x; \theta) = -\log p_\theta(y \mid x)$ and there exists some $\theta^* = \theta_\infty$ such that $p_{\theta^*}(y \mid x) = p(y \mid x)$ then*

$$\log(L(n, q)) = -\log(n) + \log\left(Tr\left(\Sigma^*\left(\sum_k q_k \Sigma_k\right)^{-1}\right) + o(n^{-1})\right).$$

*for $\Sigma_k := \mathbb{E}_{p_k}[H\ell(y, x; \theta^*)]$ and $\Sigma^* := \mathbb{E}_{p^*}[H\ell(y, x; \theta^*)]$*

**Proof**  The statement follows almost definitionally. By the conditions of the corollary statement, the model is a well-specified maximum likelihood estimator, and the fisher information and hessian coincide, $\Sigma_{\theta_\infty} = I_{\theta_\infty}$. Simplifying the expression in Corollary A.1 and expanding $\Sigma_{\theta_\infty}$ into its $k$ components gives the desired result.  $\square$

**Proposition 3.2.** *Let $B_\delta(x, p_k) = \mathbb{E}_{x' \sim p_k}[|B_\delta(x, \{x'\})|]$ be the probability of drawing some $x' \sim p_k$ in the same bin as $x$, and assume $B_\delta(x, p_k)$ is bounded away from zero. Then*

$$L(n, q) := \mathbb{E}[\ell(\hat{f}_\delta(x), x) - \ell(f(x), x)]$$
$$= \mathbb{E}\left[\frac{\ell''(f(x), x)}{\sum_k q_k B_\delta(x, p_k)}\right] + O\left(\frac{\sqrt{\log(\gamma^{-1}) + d\log(\delta)}}{\sqrt{2n}}\right) + O(L\delta\sqrt{d} + L^2\delta^2 d),$$

*holds with probability at least $1 - \gamma$, where the expectation is taken with respect to draws of $y$.*

**Proof**  Note that by definition of $y$, whenever $|B_\delta(x, X_n)| > 0$, $\hat{f}_\delta(x)$ has mean close to $f(x)$ with the deviation controlled by the Lipschitz constant of $f$,

$$\mathbb{E}_{y|x}[\hat{f}_\delta(x)] = f(x) + O(L\delta\sqrt{d})$$

and variance (considering $|B_\delta(x, X_n)|$ fixed),

$$\mathrm{Var}_{y|x}[\hat{f}_\delta(x)] = \frac{1 + \mathrm{Var}[f(x) \mid x \in B_\delta(x, X_n)]}{|B_\delta(x, X_n)|} = \frac{1 + O(L\delta\sqrt{d})}{|B_\delta(x, X_n)|}.$$

Taking the second order Taylor approximation to $\ell$ at $f(x)$ we get

$$\mathbb{E}[\ell(\hat{f}_\delta(x), x)] - \mathbb{E}[\ell(f(x), x)]$$
$$= \mathbb{E}[\ell'(f(x), x)(\hat{f}_\delta(x) - f(x))] + \mathbb{E}[\ell''(f(x), x)(f(x) - \hat{f}_\delta(x))^2/2] + o(\mathbb{E}[(f(x) - \hat{f}_\delta(x))^2])$$
$$= O(L\delta\sqrt{d} + L^2\delta^2 d) + \mathbb{E}[\ell''(f(x), x)\mathrm{Var}_{y|x}[\hat{f}_\delta(x)]] + o(\mathbb{E}[(f(x) - \hat{f}_\delta(x))^2])$$
$$= \mathbb{E}\left[\ell''(f(x), x)|B_\delta(x, X_n)|^{-1}/2\right] + O(L\delta\sqrt{d} + L^2\delta^2 d).$$

The third line follows from the expectation bound, as well as applying the bias-variance decomposition to the second order term. The last line follows from the variance identity above, and applying the same bias-variance decomposition on the $o(\mathbb{E}[(f(x) - \hat{f}_\delta(x))^2])$ term.

As with the linear regression case, we cannot simply take expectations as there is a small but nonzero probability that $|B_\delta(x, X_n)|$ is zero. We will show this happens with low probability. By Hoeffding's inequality, each of the $\delta^{-d}$ bins will concentrate towards their expected values

$$P\left(\left||B_\delta(x, X_n)|n^{-1} - \mathbb{E}_{x \sim p}[B_\delta(x, \{x'\})]\right| > \frac{\sqrt{\log(2\gamma^{-1}\delta^{-d})}}{\sqrt{2n}}\right) \leq 1 - \delta^d \gamma.$$

Applying the union bound, we have concentration at the same rate over all $\delta^{-d}$ bins with probability $1 - \gamma$.

Now note that we can write

$$\mathbb{E}\left[\ell''(f(x), x)|B_\delta(x, X_n)|^{-1}\right] = \frac{1}{n}\mathbb{E}\left[\ell''(f(x), x)\frac{1}{\mathbb{E}[B_\delta(x, \{x'\})] + |B_\delta(x, X_n)|n^{-1} - \mathbb{E}[B_\delta(x, \{x'\})]}\right].$$

We can take the Taylor approximation of the ratio at the expectation, which gives us

$$\mathbb{E}\left[\ell''(f(x), x)|B_\delta(x, X_n)|^{-1}\right] = \frac{1}{n}\mathbb{E}\left[\ell''(f(x), x)\frac{1}{\mathbb{E}[B_\delta(x, \{x'\})]} + O(|B_\delta(x, X_n)|n^{-1} - \mathbb{E}[B_\delta(x, \{x'\})])\right].$$

Now with probability at least $1 - \gamma$,

$$\mathbb{E}\left[\ell''(f(x), x)|B_\delta(x, X_n)|^{-1}\right] = \frac{1}{n}\mathbb{E}\left[\ell''(f(x), x)\frac{1}{\mathbb{E}[B_\delta(x, \{x'\})]}\right] + O\left(\frac{\sqrt{\log(\gamma^{-1}) + d\log(\delta)}}{\sqrt{2n}}\right).$$

Plugging this into our earlier expression, and expanding the expectation in terms of each of the data sources completes the proof.

$\square$

