# OpenReview forum: "Predicting the impact of dataset composition on model performance"
_ICLR.cc/2021/Conference — Reject_

### Official Review · AnonReviewer3 · 2020-10-14
**A neat result in theory; Practical use is in question due to the unclarity of how to compute excess loss without knowing the oracle**

**Rating:** 5
**Confidence:** 4

**Review:**

This work studies the problem of predicting model performance with more training data when the data are collected from different sources. The predictor is a function of the number training examples, and the ratio of examples from each source. The predictor needs to be built from a small number of training examples the observed model performance, and applied to larger numbers of training example without actually training the model. The predictor can be used to decide a good data collection policy which is expected to have best model performance. The proposed solution is a simple parametric form of the predictor, which is log-linear in the log of # training examples, and log-rational in the source distribution vector. The solution is motivated by recent literature about the same task for single source. The correctness of the solution is proved for several cases: linear regression, M-estimator and nonparametric binning. The performance of the predictor is then evaluated for a number of real-world tasks: linear regression for Amazon book rating, semantic parsing, machine translation and multitask question answering. The performance is measured by r2 score between the actual performance and predicted performance. The proposed predictor has a clear advantage over the baseline of using a linear predictor.

Pros:

1. The work is built on top of recent studies with single source and the extension to multi source is timely.
2. The discovered relation between the expected model performance and the number of training examples from multiple sources is neat and interesting.
3. The proposed rational function of the distribution vector is justified for a few cases theoretically.
4. The empirical evaluation uses diverse real tasks, covering different loss functions, including the ones where the assumptions in the theoretical analysis are not satisfied. And the proposed solution has a reasonably robust performance and is much better than the baseline. The behavior of the predictor across the tasks meets expectation.

Cons:
1. " Following earlier observations by Hestness et al. (2017), we expect a log-linear relationship
between L(n, q) and log(n) for any fixed q," It will be good to specify the condition of the loss function required by the reference.
2. The experiment applies prediction to 2-6$\times$ of the training examples used for fitting the predictor at max. It will be useful to study the case when this gap is larger, to understand the power or limitation of the proposed predictor.
3. The distribution vector is randomly sampled in both training and testing. It will be useful to test the case where the distribution vector is sampled differently, to further evaluate the extrapolation ability.
4. Combining 2 and 3, it will be very useful to test the limit of the proposed approach and know the minimal required observations for the predictor to work well.
5. Page 8: typo BLUE.

===================
Update after discussion: The difference between excess loss and absolute loss is an issue I overlooked. After discussing with other reviewers, I realized a new question: how to compute the excess loss L(n, q) without knowing the the best possible estimator (oracle) in the class? This is a key step in the proposed algorithm when fitting the predictor. The paper has not explained it. As of now I do not think that is possible, except if strong assumptions are made, such as the oracle has 0 loss. In the experiments, if the authors use the estimator trained with full data as the oracle to compute the excess loss, that would invalidate the practical usability of the approach because the goal was to predict the performance without having full data to begin with.

---

> ### Author Response · Authors · 2020-11-20
> **Responses to comments**
>
> 1. Hestness: Thanks for the comment - I believe Hestness is mainly empirical and showed this relationship for classification error, cross-entropy, and word-error-rate. We'll clarify this in an update.
> 2. Extrapolation: This is a good point and we'll update this if able, but we will have to find some new data or do computationally intensive MT model training to achieve this, as we need some basic amount of data to train a model with small n, and scaling up 10-20x makes the datasets used for evaluation large. I believe similar constraints apply to existing scaling law papers and thus they also do not try to search for the large-n breakdown point.
> 3. We answer this in another rebuttal as well, but our current approach does not handle extreme extrapolation on q (near 0 or 1) particuarly well and we're exploring regularization methods to try to get this to work.
> 4. BLUE: this is an embarrassing typo, and I'm thankful that you caught this.

---

> > ### Comment · AnonReviewer3 · 2020-11-24
> > **Followup question on the difficulty of extrapolation study**
> >
> > Thanks for the discussion on the extrapolation study. I hope to get more clarity about why it is difficult to do this study with the current data. In theory, you can use the current dataset but begin with a small n (e.g., 60K instead of 600K). If there is a reason that cannot be done for any of the tasks studied, I'd be interested to hear. Otherwise, showing some experiment results in that setting, even if negative, would help readers understand the limitation. Also, if there is a 'basic amount of data' requirement by the proposed approach, making this requirement clear and having guidelines about determining the minimal required amount of data will be helpful.

---

> > > ### Author Response · Authors · 2020-11-25
> > > **Notes and a quick experiment.**
> > >
> > > I agree with your point that we could always reduce the number of training samples to hit the desired extrapolation ratio.
> > > We'll first present some experiments with very low sample size -> large extrapolation that we were able to run today, and then afterwards we'll discuss limitations and why we didnt initially focus on this.
> > >
> > > We ran two quick experiments on our smaller datasets, as this required no model hyperparameter tuning or large neural net training. On amazon, we increased the ratio to 10-1 by subsampling the training data down to a somewhat extreme ~100 examples per category. This results in a r^2 of 0.75 for the model (for reference, linear is 0.64 and datasize is 0.31). This is worse, and we think it's because with so few examples the stochasticity from the sampled examples and non-asymptotic behavior of the model dominates.
> > >
> > > We also set up a case where we extrapolate by increasing the total data size. In the toy regression data setting, we can generate as much data as we want and we get 10-1 extrapolation (from 500 to 6000 samples) with ~ 0.99 r^2.
> > >
> > > To discuss why we were interested in adding (rather than removing data): In the submission and rebuttal, we were focused on the case where the user has collected enough data to get a usable system of some kind - this was the basis of how we designed the datasizes for the 3 experiments. Our main interest was whether we could extrapolate out from this 'minimal working example' setting to the large scale data collection case, rather than in asking whether we could extrapolate the performance of a model from a few-shot like setting. This drove the design of the approach, which is based on asymptotics of excess error. Reading your rebuttal, I agree that it's still interesting to investigate this lower data extrapolation regime, and we'll be adding this result as well as a more careful discussion of our assumptions (enough data to get asymptotics) to the paper.

---

### Official Review · AnonReviewer1 · 2020-10-27
**Interesting and important question; experimental weaknesses**

**Rating:** 7
**Confidence:** 4

**Review:**

This paper casts generalization under covariate shift as a prediction problem. Concretely, given a model trained on $n$ samples from a mixture of $K$ distributions, can we predict the excess risk of the model on data drawn from a different mixture?
Motivated by recent empirical work, the paper posits a simple functional form for the model: log-linear dependence on dataset size and an additive offset given by a \emph{rational} function of the mixture coefficients. The author's demonstrate theoretically that this form arises naturally for linear regression, M-estimation, and a non-parametric binning estimator. Empirically, they show fitting this model to data gives better predictive performance than methods based on dataset size or using linear functional forms.

Pros:
- This paper addresses a question of great interest to much of the ML community: generalization under distribution shift.
- The approach taken here, i.e. using theory to posit "scaling laws" or models whose parameters are fit to then fit to empirical data is an interesting alternative to reasoning about performance under distribution shift from first principles alone.
- The quantitative predictions from the models presented here may be of interest to people using active learning to gather new data to improve model performance.
- The paper and proofs read well and don't appear to have correctness issues. (I did not carefully verify Proposition 3.2 due to time).
- In the experiments presented, the rational function model gives fairly accurate predictions, and certainly better predictions than methods based on dataset size alone.

Cons:
- My main complaints with this paper is the lack of limited baselines and the limited scope of the experimental results.
- Baselines: The dataset size and linear baselines are good starting points, but the the experiments not include any non-linear baselines. This makes it difficult to determine whether the rational function model is really fundamental or necessary. Would, e.g., a generic function approximator perform just as well?
- Experimental scope: Scaling laws like the ones presented in this paper are useful and interesting in so far as they capture some "universal" phenomenon.
-- In each of the settings considered, the experiments focus on one model, e.g. pretrained BERT for question answering. How general is this method beyond the models considered? Does the rational model give predictive power on each task across a diverse set of models, or is there something particular about the ones presented?
-- All of the experiments take place with text data. Does a similar result hold, e.g., for CNNs trained with natural images?

Overall, I very much enjoyed reading this paper, and it offers several avenues for further inquiry. I vote to accept.

==============

Update after rebuttal:
Thanks to the authors for their response. I enjoyed this paper, and I'm keeping my score unchanged. In terms of multiple datasets/diverse models, I appreciate the various models/dataset/baselines currently included as a proof of concept. However, I'd be very interested in a more systematic study with significantly more models and datasets to understand better precisely when excess error as function of data composition and size can be reliably extrapolated and the extent to which the trends observed are "universal."

---

> ### Author Response · Authors · 2020-11-20
> **Baselines and evaluations**
>
> 1. Baselines: We've added a MLP baseline that hopefully helps answer the question of whether an overparametrized function approximator works. For this set of experiments, we wanted to focus on multiple domains rather than multiple baselines, much like Hestness 2017 who studied many tasks and models but only considered a single power-law functional form. We agree that it would be useful to add more baselines (as we did with the MLP) but this paper is still valuable in showing that a very simple functional form *can* predict the excess error of models as a function of data composition and size - the fact that a more complex model can also do this is interesting, but we don't think should be grounds for rejecting this paper.
> 2. Re diverse models: this is exactly why we picked different models for each setting (ridge for amazon, LSTM-based semantic parser for dialogue, transformers for MT, BERT fine-tuning for QA) so that we can give a sense of how well this works across models and tasks.
> 3. CNNs: This would be something we'd like to do, though it's unlikely to fit within the time of the rebuttal. We initially focused on language tasks, as we were substantially more familiar with multi-domain settings and models.

---

### Official Review · AnonReviewer2 · 2020-10-27
**Good paper, but I am not convinced that L(n, q) is separable in n and q.**

**Rating:** 5
**Confidence:** 3

**Review:**

In this paper, the authors proposed a model to predict the model performance given the sample size n and the composition of the source of the data, q. They argued that the excess test loss should be separable in n and q.  The paper is interesting and well presented. I have no difficulty following the authors arguments. The proposal could be useful for some practitioners.

However, I am not convinced that log(L(n, q)) should be separable in n and q in general. The authors did show in some special cases in Figure 1 and Sections 3.1~3.3 that n and q are separable, but a more general theory is lacking. The baselines in the experiments are all relatively weak. In fact, comparing the results between Datasize and Rational, it seems plausible to me that Rational achieves good performance simply due to the added flexibility in the q term, rather than modeling the n and q relationship correctly. A more suitable baseline would be to use a DNN model on [n, q], without separating n and q, and if Rational achieves on par performance with DNN on Train, that suggests to me that the functional form of Rational is probably correct.

On page 2, the authors wrote "The performance of this classifier is evaluated on a test distribution which may differ from the
training distribution by a covariate shift (i.e., p(y|x) = p_test(y|x))." Is there a typo here? Do the authors want to write "p(y|x) = p_test(y|x) but P(x) \neq p_test(x)"?

On page 3, what is the expectation taken over in L(n, q)?

On page 3, I can only see 3 colors in Figure 1 (a), yet there are 5 lines in the legend.

On page 4, in the formulation of C(q), lambda appears in the denominator. How to initialize lambda such that sum(lambda q) is not too small, leading to exploding C(q)? Is there any constraint on alpha to guarantee the sublinear relationship established in Hestness et al. (2017)? Is there any constraint on lambda's? How many lambda's should we use? It seems that based on the propositions, there should optimally be dk lambdas. This seems like a large number.

On page 7, in Table 1, do we extrapolate only with respect to n? If so, it would be great to also present a study with extrapolation on q.

---

> ### Author Response · Authors · 2020-11-20
> **Replies and a reference to MLP experiments**
>
> 1. Separability: Thank you for the comment - we agree this was a gap in the paper and we've added a new non-separable MLP experiment as a comment to all reviewers. Let us know if there are further questions.
> 2. Page 2 covariate shift: We agree this is unclear. Technically the "may differ" catches the case of p(x)=p_test(x) but this is not really a case we care about, so we'll remove that and clarify.
> 3. Expectation on L: This is over draws of the training and test distributions. We'll make the description more explicit before we define L(n,q)
> 4. Page 4: We use a log-exp-parametrization, which we've found to be stable. The objective (page 4) is log(C(q)) with C(q) = sum (sum lambda q)^{-1} . We parametrize log C(logq) = -log(sum( sum lambda exp(logq))).  This can still blow up if logq → -inf, but we have not experienced this for any reasonable step size settings (< 0.01).
> 5. Hestness constraint: we don't enforce that |alpha|< 1 , since we found that this would happen naturally in our experiments.
> 6. Number of lambdas: this is still a somewhat open question. We found this to be pretty stable - in initial experiments for amazon, we found M=2 to 8 produced qualitatively similar results. The 'true' number of lambdas is very large, but I think one reason our paper is interesting is that in practice, it seems like a small M and the diagonal approximation are both effective in practice.
> 7. Yes, we extrapolate only with respect to n. Extrapolation with respect to q is alot more challenging and might be out of scope for this paper. We're currently experimenting with some regularization on q with the hope of making this feasible, and will include this if time permits.

---

### Official Review · AnonReviewer4 · 2020-10-28
**Interesting paper, but lacks clarity**

**Rating:** 4
**Confidence:** 4

**Review:**

### Summary of the contribution
The paper addresses the issue of combining datasets from different sources, and taking into account the fact that they do not share the same precision.
It aims at predicting the performance of an algorithm trained on those data, through a simple formula taking into account the size of the data, and its composition through the mixture weights of the distribution from which it comes.
As the formula also depends on the estimator trained and the quality of data, it is trained to be estimated fully.

### Strengths
The paper and the topic are both interesting. The maths look correct, although I have not checked it all.

### Weaknesses and concerns
The paper lacks clarity in general.
1. Context
The context of mixed-source data is not very clear:
	- from Section 2, it appears to be applicable e.g. for datasets coming from different sensors, but predicting the same output, while the experiment section makes use of very different datasets, not even predicting the same target;
I encouraged the authors to provide specific examples in Introduction or Section 2 for clarity. The experiments seem to be related to multitask learning, as mentioned shortly for one application but looks to be true also for others;
it could also possibly be related to domain adaptation. In both cases, the experiments should include comparison to state-of-the-art approaches in those learning problems.
	- It is also not clear how the different sources are mixed together: does the proportion $q$ apply to the number of observations in each source?
	- Moreover, there is mention in the Introduction of the goal of optimal data collection policy.
How can the proposed work be used in practice for that goal, since we do not know in advance what is the quality of each dataset, or how well does an estimator perform?
2. Main goal of the work
The title and most of the work is a bit misleading, as it mentions predicting the performances of an estimator, while in fact it predicts its excess loss, that is, how close an estimator is from the best possible estimator (oracle) in the model;
while this is a good approach to improve the fitting of an estimator (and its parameters) inside a family, it does not give any clue as its actual performance.
It is also not clear what the learning problem at the beginning of Section 3 does, and how it is used in practice.
3. Related work
Appart from the learning problems mentioned before, here are possible issues with related work
	- Active learning does not qualify as related work, as it is usually concerned with the labelization of unlabeled samples in a single source dataset when labelling is expensive.
If there exist work on that domain that is specific to the problem of combining data from different sources, please provide a more specific reference.
	- There exist however other works for the issue of performance evaluation through a learning problem, e.g. "Per Instance Algorithm Configuration of CMA-ES with Limited Budget" by Belkhir et al (2017);
 it should be interesting to compare to the proposed work.
4. Experiments
The experiments show the following issues:
	- As the proposed formula estimates the excess loss and not the performance itself, there should also be a comparison of the actual performances obtained both in the full training set and on the test set.
	- Amazon sentiment:
		- regression is applied on the sentiment prediction, which is a classification task
		- the estimator considered is ridge regression, which does not fit Proposition 3.1 setting and no comments are made as to how the penalty would impact (or not) the theoretical properties of the work
		- $C(q)$ contains 4 terms, while it said that there are 3 data sources in training

### Minor comments:
Proposition 3.2 is not written in the same format as the others, and makes it hard to see the link with n (or log n). Please re-write it or explain where the n component is.

### Overall evaluation:
The paper is interesting and tackles an important problem. However, it lacks clarity in many aspects.

=====POST-REBUTTAL COMMENTS========

I thank the authors for answering my questions. Their answers did clarify some aspects, such as the ratio of mixed sources, the optimal data collection, and the relation of the proposed work to active learning and multitask. However, the answer provided on my comment about the distinction between excess loss and absolute loss is not sufficient. I believe this is an important point, and calls for a major revision, not just a promise to clarify the point throughout the paper. The same comment applies to the experiment when comparing methods on the basis of computing the excess loss and not showing the absolute loss, which is the actual measure of quality of an estimator.

---

> ### Author Response · Authors · 2020-11-20
> **Thank you for your comments**
>
> Thank you for your notes on clarity and presentation. We will include the following clarification points into the draft:
>
> 1. We will include concrete examples in the intro - we will use the amazon sentiment task (3 different types of reviews, goal is to have a system that performs well on book reviews) in the intro to motivate the problem.
> 2. Re multitask/domain adaptation: the system we use for BoolQ (the only multitask setting) is a multitask learning baseline (the Jiant package). We will clarify this in the update. Regarding SoTA methods for multitask/multi-domain, we tried using a near SoTA multi-domain MT system (Hu 2019) early in the project but found the computational burdens to be a problem. If time permits, we'd like to revisit this and set up and run a SoTA MT baseline.
> 3. The proportion q_i controls the fraction of data coming from data source i.
> 4. For optimizing data collection, our argument is that one could collect a small, cheap pilot dataset and learn how q impacts excess error. Using this, it's possible to pick a q that optimizes excess error for a given data collection budget. This is consistent with our experiment setup where we fit L(n,q) on a small dataset and evaluate performance on extrapolation for large n. We will include this argument more explicitly.
> 5. Main goal / excess loss: thank you for this point, we were unclear about the distinction between excess / absolute error and we will make this explicit from the very start. To your point about absolute vs excess loss and the value of our procedure: we think scaling laws for excess loss is interesting and valuable. The existing literature of model and data scaling laws have the same limitation, but have had a substantial impact in terms of the design of models and data collection (Hestness 2017, Kaplan 2020, Rosenfeld 2020) since many situations involve a fixed model class (e.g., convnets or transformer models).
> 6. Learning problem: we will make this clearer - our approach produces a model for L(n,q) given a small pilot dataset, and the learning problem in section 3 is the learning procedure for L(n,q).
> 7. Active learning: we included active learning as a reference since many active learning algorithms involve implicit measures of how data collection impacts downstream performance - many data valuation papers such as Ghorbani 2019 make this explicit. We will clarify this further, and make the references to data valuation papers more explicit.
> 8. Belkhir: thank you for this cite. This brings up a point that response-surface methods and other related ideas in black box optimization share a similar motivation in using surrogate functions; we will include this in a revision.
> 9. Experiments - excess vs absolute loss: note that the experiments evaluate r^2, which is invariant to adding the bayes error to both the predicted and true errors. We will include the absolute errors achieved by all methods into the paper with a note.
> 10. Experiments - amazon: please note that we use Mansour et al's version of the amazon sentiment prediction task, which is a regression task of predicting the number of stars from 1-5.
> 11. Experiments - ridge: Part of the purpose of the experiments is to validate that our predictions hold outside the exact theory conditions. In the case of ridge regression, Prop 3.1 can be updated to use the ridge regression coefficients beta instead of the OLS coefficients, and the solution will involve a covariance matrix with shrinkage.
> 12. The number of degrees of freedom (M) is a separate parameter from the number of data sources (K). To minimize the amount of hyperparameter tuning, we fixed this to M=K+1 in all experiments, to be able to at least model the case where each data source contributes independently.
> 13. 3.2 - thank you for this catch. We've fixed this typo - there's a 1/n factor in front of the expectation.

---

### Author Response · Authors · 2020-11-17
**New MLP baseline experiments**

Thank you for the in-depth and thoughtful comments. We will address all the comments in turn but wanted to post additional experimental results now to address a comment that required additional experiments (these experiments will be included in a revision to come).

Two reviewers commented upon the baselines used in the paper and suggested the use of non-linear or neural network baselines. In particular, there is a question about whether any flexible function approximation would perform just as well as the 'rational' approximation.

This turns out not to be the case. We trained 3 different fully-connected multilayer neural networks: mlp-overparam (5 hidden layers, 50 neurons each), mlp (layers and width both set to num_data_source+1), mlp-onelayer (1 layer and width set to num_data_source+1).

1). MLP-overparam tests the hypothesis that any flexible function approximator succeeds, and we picked the hyperparams to achieve 0.99 train r-squared across the tasks.

2). MLP tests the performance of a model whose parameter count scales with the number of data sources, but is still overparametrized.

3). MLP-onelayer tests the performance of MLP when its number of parameters is comparable to the rational approximation.

The performance of MLP models are below (r-squared for rational repeated here for ease of comparison)

Amazon extrapolation:

0.96 rational, 0.91 mlp, -0.50 mlp-onelayer, 0.33 mlp-overparam

Amazon train:

0.95 rational, 0.92 mlp, 0.86 mlp-onelayer, 0.99 mlp-overparam

Dialogue extrapolation:

0.93 rational, -0.4 mlp, -4.9 mlp-lowparam, -8.5 mlp-overaparam

Dialgoue train:

0.99 rational, 0.99 mlp, 0.95 mlp-lowparam, 0.99 mlp-overparam

MT extrapolation:

0.82 rational, -0.95 mlp, -1.1 mlp-lowparam, -1.9 mlp-overparam

MT train:

0.97 rational, 0.91 mlp, 0.88 mlp-lowparam, 0.99 mlp-overparam

QA extrapolation:

0.86 rational, 0.13 mlp, 0.56 mlp-lowparam, -0.34 mlp-overparam

QA train:

0.98 rational, 0.99 mlp, 0.98 mlp-lowparam, 0.99 mlp-overparam

Extrapolation performance is strongly in favor of the rational approximation. The overparametrized models even underperform linear models in some cases, due to the fact that not decoupling [n, q] makes learning and extrapolation very challenging.  This is also not due to parameter counts either too large or small relative to rational, since mlp-lowparam and mlp-overparam cover both ends of the spectrum.

Train performance is what you'd expect, with overparam essentially interpolating the data. This does not provide evidence for MLP being useful, as it takes multiple orders-of-magnitude more parameters to match rational's fit on the training data. Only amazon train shows any gap between rational and mlp-overparam and the substantially worse extrapolation performance of mlp-overparam suggests overfitting.

---

### Decision · Program_Chairs · 2021-01-07
**Final Decision**

**Decision:**

Reject

**Comment:**

This paper studies the following broad question: How can we predict model performance when the data comes from different sources? The reviewers agreed that the direction studied is very interesting. While the results presented in this work are promising, several reviewers pointed out some weaknesses in the paper, including a confusion between absolute loss and excess loss, and the limited scope of the experiments. Overall, this paper does not appear to be ready for publication in its current form. In my personal opinion, if the concerns raised by the reviewers are appropriately addressed, this work could be publishable in a high quality venue.